

# Soil metabolomics and bacterial functional traits revealed the responses of rhizosphere soil bacterial community to long-term continuous cropping of Tibetan barley

Yuan Zhao[1,*], Youhua Yao[2,*], Hongyan Xu[1,2], Zhanling Xie[1], Jing Guo[1], Zhifan Qi[1] and Hongchen Jiang[3]

[1] Qinghai University, College of Eco-Environmental Engineering, Xining, Qinghai, China
[2] Qinghai University, Academy of Agriculture and Forestry Sciences, Xining, Qinghai, China
[3] China University of Geosciences, State Key Laboratory of Biogeology and Environmental Geology, Wuhan, Hubei, China
[*] These authors contributed equally to this work.

Corresponding author
Zhanling Xie,
xiezhanling2012@126.com

## ABSTRACT

Continuous cropping often leads to an unbalanced soil microbial community, which in turn negatively affects soil functions. However, systematic research of how these effects impact the bacterial composition, microbial functional traits, and soil metabolites is lacking. In the present study, the rhizosphere soil samples of Tibetan barley continuously monocropped for 2 (CCY02), 5 (CCY05), and 10 (CCY10) years were collected. By utilizing 16S high-throughput sequencing, untargeted metabolomes, and quantitative microbial element cycling smart chips, we examined the bacterial community structure, soil metabolites, and bacterial functional gene abundances, respectively. We found that bacterial richness (based on Chao1 and Phylogenetic Diversity [PD] indices) was significantly higher in CCY02 and CCY10 than in CCY05. As per principal component analysis (PCA), samples from the continuous monocropping year tended to share more similar species compositions and soil metabolites, and exhibited distinct patterns over time. The results of the Procrustes analysis indicated that alterations in the soil metabolic profiles and bacterial functional genes after long-term continuous cropping were mainly mediated by soil microbial communities ($P < 0.05$). Moreover, 14 genera mainly contributed to the sample dissimilarities. Of these, five genera were identified as the dominant shared taxa, including *Blastococcus*, *Nocardioides*, *Sphingomonas*, *Bacillus*, and *Solirubrobacter*. The continuous cropping of Tibetan barley significantly increased the abundances of genes related to C-degradation ($F = 9.25$, $P = 0.01$) and P-cycling ($F = 5.35$, $P = 0.03$). N-cycling significantly negatively correlated with bacterial diversity ($r = -0.71$, $P = 0.01$). The co-occurrence network analysis revealed that nine hub genera correlated with most of the functional genes and a hub taxon, Desulfuromonadales, mainly co-occurred with the metabolites via both negative and positive correlations. Collectively, our findings indicated that continuous cropping significantly altered the bacterial community structure, functioning of rhizosphere soils, and soil metabolites, thereby providing a comprehensive understanding of the effects of the long-term continuous cropping of Tibetan barley.

## INTRODUCTION

Soil health sustains the capacity of agricultural soils for ecosystem functioning and thrives as a living ecosystem for microbes, plants, and insects (*Lal, 2016*; *Rinot et al., 2019*). However, continuous cropping, especially of a monoculture consisting of one plant type in the same agricultural field with standard and persistent agronomic practices, usually leads to crop yield reductions, soil aggregation, soil physicochemical property alterations, and soil microbial community changes, ultimately exerting negative effects on soil health (*Murphy & Lemerle, 2006*; *Pervaiz et al., 2020*). Soil microbiota play crucial roles in the maintenance of soil functions and prominently affect agricultural soil productivity, plant growth, and crop quality (*Bello-Akinosho et al., 2016*; *Huang et al., 2013*; *Pii et al., 2016*). Previous studies demonstrated that continuous monoculture greatly affected the structure of the rhizosphere soil microbial community. For example, successive monoculture practices were found to decrease bacterial community diversity (*Sun et al., 2018*), increase harmful populations, and inhibit beneficial ones (*Gao et al., 2019*). In our previous study, we found that the continuous cropping of Tibetan barley resulted in a significant decline in crop yields and bacterial community diversity, as well as increased the relative abundances of bacteria associated with chemo-heterotrophy, aromatic compound degradation, and nitrate reduction, per a FaProTax prediction (*Yao et al., 2020*). However, research on how continuous cropping practices affect soil microbial functional traits is lacking.

Nutrient cycling, including carbon (C), nitrogen (N), and phosphorus (P), is a common problem that has been investigated on a global scale (*Zhang et al., 2019*). The tight interactions in the rhizosphere zone between plants and soil microbiota affect the stability of nutrient cycling in the soil (*Zhang et al., 2019*). For example, *Kuzyakov (2010)* reported that soil microbial activity was enhanced by a substantial source of soil C from root secretion and deposition. *Yu et al. (2019)* found that the activities of key functional genes within the microbial community involved in N-cycling increased along with a reduction in N application during maize/soybean strip intercropping. *Alami et al. (2020)* found that the abundances of soil N functional genes significantly differed between cultivated and fallow fields across two seasons based on a PICRUSt prediction. *Pang et al. (2021)* indicated that the continuous cropping of sugarcane significantly decreased the bacterial abundances associated with rhizosphere soil N- and sulfur (S)-cycling, thereby decreasing the abundances of N translocation genes and dissimilatory reduction genes, as determined by the soil metagenome. Unfortunately, less is known about soil functional gene cycling and their interactions with microbial communities underlying the continuous cropping of Tibetan barley.

Most studies on continuous cropping have generally focused on changes in the microbial community and their interactions with soil properties, while soil metabolite composition is rarely discussed. The rhizosphere, which is a hub of microbial activities,

increases the nutrient supply for microorganisms, as the roots release several organic compounds that influence plant growth and health (*Pinton, Varanini & Nannipieri, 2007*). Thus, soil metabolomics can potentially enhance our understanding of this chemical exchange. Fortunately, the development of untargeted metabolomics has allowed us to detect and identify increasingly more compounds that are secreted by plants and the organisms that interact with them in the rhizosphere (*Swenson et al., 2015*; *Withers et al., 2020*). For example, *Wang et al. (2020)* studied the continuous cropping of ramie by combining rhizosphere microbe identification and non-targeted gas chromatograph-mass spectrometer (GS/MS) metabolome analysis, and found that bacteria, such as Rhizobia, synthesized IAA and likely reduced the biotic stress of ramie. Nevertheless, the contribution of the changes in soil metabolites, their co-occurrence network, and microbial composition on the long-term monoculture of crops are far less understood, especially in Tibetan barley.

Thus, in the present study, we collected the rhizosphere soils of Tibetan barley after two short-term and one long-term periods of cropping. By applying high-throughput sequencing and liquid chromatography mass spectrometry (LC-MS) untargeted metabolomics, we investigated the change trends in soil bacterial community structures and differences in soil metabolites. By constructing a co-occurrence network, we determined the changes in the interactions of bacterial communities. Quantitative microbial element cycling (QMEC) was applied to detect the abundances of C-, N-, P-, and S-cycling-related genes in the rhizosphere soils of different continuous cropping years (*Zheng et al., 2018*). Our findings will enhance our understanding of the effects of the long-term continuous cropping of Tibetan barley on the direct interactions between specific functional taxa and important functional gene metabolism, and provide a comprehensive understanding of the effects of the long-term continuous cropping of Tibetan barley.

## MATERIALS & METHODS

### Site description and rhizosphere soil sampling

The study was carried out at an alpine Tibetan barley experiment site located in the Qinghai–Tibetan Plateau (37°21′N, 101°43′E; altitude, 3,700 m), Beishan Township, Menyuan County, Qinghai Province, China. This region has a plateau continental climate with a mean annual temperature of 1.3 °C and mean annual precipitation of 530–560 mm. According to the Food and Agriculture Organization (FAO) classification, the soil type is classified as Kastanozems (*Schad & Spaargaren, 2006*). Kunlun 14 barley (*Hordeum vulgare* L.) was used as the experiment material. The study site consisted of independent plots over many years of the successive monoculture of Tibetan barley. Tibetan barley seeds were cultivated in April each year after the annual application of blended fertilizer and harvested in August following previously described methods (*Yao et al., 2020*). Briefly, in July 2020, we selected three 5 m × 5 m quadrats including 2, 5, and 10 years of successive monocropping of Tibetan barley, which we named CCY02, CCY05, and CCY10, respectively. Each quadrat was originally divided into five rectangular 5 m × 1 m sub-quadrats. In each sub-quadrat, we collected a total of 15 rhizosphere soil samples (closely adhered to the roots) in a "Z" pattern and all rhizosphere soil samples within a quadrat were mixed together as a

single biological sample. The same steps were repeated four times to make four biological rhizosphere soil duplicates per continuous monocropping year. Finally, we obtained a total 12 rhizosphere soil samples for the subsequent experiments. Soil samples were then immediately transferred on ice to the laboratory and stored at −20 °C for subsequent analysis.

## High-throughput sequencing of bacterial 16S rRNA gene

The genomic DNA was extracted from each rhizosphere soil sample using an E.Z.N.A®. Soil DNA kit (Omega Bio-Tek, Norcross, GA, USA) following the manufacturer's instructions. The bacterial V3-4 region was then amplified with the primer set 338F (5′-ACT CCT ACG GGA GGC AGC AG-3′) and 806R (5′-GGA CTA CHV GGG TWT CTA AT-3′) (*Zeng & An, 2021*). PCR amplification and purification were performed following previously described methods (*Yang, Liu & Zhang, 2019*). Briefly, each sample consisted of a 30 µL mixture in triplicate, comprised of 10 µL TaKaTa EX Tag PCR premix (TaKaRa Bio Inc., Kusatsu, Japan), 0.5 µL of each primer (10 µM), 1 µL DNA template (20 ng), and 18 µL PCR-grade water, which was amplified according to the stetted procedure (95 °C for 5 min, followed by 25 cycles at 95 °C for 30 s, 55 °C for 30 s, 72 °C for 30 s, and a final extension at 72 °C for 10 min). The replicate PCR reactions for each triplicate were pooled and purified using a QIAquick Gel Extraction Kit (Qiagen, Chatsworth, CA, USA). A single composite sample for sequencing was prepared by combining approximately equimolar amounts of PCR products from each sample. The prepared PCR products were then paired-end sequenced (2 × 250 bp) on an Illumina MiSeq platform (Majorbio Biotech CO., Ltd., Shanghai, China) following the manufacturer's instructions.

## Soil non-targeted metabolomic detection and analysis

Twelve rhizosphere soil samples were sent to Majorbio (Shanghai, China) on dry ice for metabolite extraction, detection, and analysis. In detail, 1,000-mg soil aliquot of each sample was homogenized with 1,000 µL methanol/water (4:1, v/v) solution, including 0.02 mg/mL internal standard (L-2-chlorophenylamine acid) for 6 min at −10 °C and 50 kHz using a Geno-grinder 2,000 (SPEX, Metuchen, NJ, USA) and spun down for 30 min at 5 °C and 40 kHz. After resting for 30 min at −20 °C, each material was centrifuged at 13,000 g (relative centrifugal force) for 15 min at 4 °C, then the supernatant was transferred and concentrated by a Termovap Sample Concentrator (DC-24, Anpel Laboratory Technologies, Shanghai, China). The dry residue was derivatized by adding 50 µL acetonitrile/water (1:1), homogenized for 30 s at 5 °C and 40 kHz, and centrifuged at 13,000 g (relative centrifugal force) for 10 min at 4 °C. Finally, the supernatant was subjected to LC-tandem MS (MS/MS) analysis; 20 µL supernatant of each sample was mixed for the quality control sample.

The UHPLC-Triple TOF system (AB Sciex, Foster City, CA, USA) equipped with an ACQUITY UPLC HSS T3 (100 mm × 2.1 mm i.d., 1.8 µm; Waters, Milford, CT, USA) was applied to chromatographic separating the metabolites with two mobile phases, phase A (95% water and 5% acetonitrile (with 0.1% formic acid)) and phase B (5% water (with 0.1% formic acid), 47.5% acetonitrile, and 47.5% isopropanol). The whole system was
integrated to a quadrupole time-of-flight mass spectrometer (Triple TOFTM 5600th; AB Sciex, Foster City, CA, USA) equipped with an electrospray ionization source and operated in positive and negative modes.

The online analysis platform (Majorbio, Shanghai, China) was applied to the raw data following the manufacturer's instructions. Briefly, the ProgenesisQI (Waters Corp., Milford, CT, USA) was used for baseline filtering, peak recognition, integration, retention time correction, and peak alignment, which produced the retention time, mass charge ratio, peak intensity, and data matrix. Then, the characteristic peak was detected and the information from the MS and MS/MS analyses was mapped with a metabolic specific database; the mass error was set to <10 PPM. Finally, the metabolites were identified according to the matching scores of the secondary MS. The data were normalized with Pareto scaling and log-transformed before further analysis.

## QMEC

Twelve qualified soil DNA samples were sent to Guangdong Magigene Biotechnology Co., Ltd. (Guangzhou, China) for QMEC analysis, wherein a high-throughput quantitative PCR (qPCR)-based chip was applied to assess the microbial functional potential, including 71 functional genes related to C-, N-, P-, and S-cycling (*Zheng et al., 2018*). QMEC manipulation was conducted following previously described methods (*Chen et al., 2020a*). Briefly, DNA templates and qPCR reagents were added to the sample source-plate; each primer and qPCR reagent was added to a separate source-plate. In each run, using a non-template reaction as the negative control, 100-nL reactions were mixed in parallel three times using an automated high-throughput sample preparation device, and then added to the nanopore of the qPCR chip on a Wafergen Smart-Chip Real-time PCR platform (Wafergen, Fremont, CA, USA). Specifically, a threshold cycle (CT) of 31 was used as the detection limit and multiple melting peaks; amplification efficiencies outside the range of 0.9–1.1 were discarded.

## Data and statistical analyses

The 16S rRNA gene raw reads were processed by the QIIME2 v2020.08 pipeline (*Bolyen et al., 2019*). Briefly, paired-end reads were joined by FLASH v1.2.11 (*Magoč & Salzberg, 2011*), then sequences were demultiplexed by the q2-demux plugin. Afterwards, plugin q2-dada2 was used to conduct quality control, chimeric sequence removal, and sequence clustering. Taxonomic analyses were conducted using the plugins q2-feature-classifier and SILVA v132 database (*Quast et al., 2012*). The $\alpha$ diversity of the bacterial communities was represented by the Chao1, Shannon, ACE, and Faith's phylogenetic diversity (PD) indices, which were calculated from the amplicon sequence variants (ASVs) table rarefied to 20,349 sequences in QIIME2 for each sample.

Beta diversity metrics differ widely in the types of differences they detect, therefore, four distance-based principal coordinate analysis (PCoA), including the Bray–Curtis, Jaccard, Weighted Unifrac, and Unweighted Unifrac were applied to visualize the differentiation among samples. Non-parametric multivariate statistical tests, Kruskal-Wallis or Adonis, were implemented to test for significant differences between the variances of bacterial

**Table 1  Significant differences of the four alpha indexes among groups.**

|  | Chao1 | ACE | PD | Shannon |
|---|---|---|---|---|
| CCY02 | 4746.71 ± 155.81a | 4825.48 ± 137.84a | 179.89 ± 7.65a | 9.69 ± 0.16ab |
| CCY05 | 4344.43 ± 195.09b | 4387.67 ± 188.26b | 162.66 ± 4.50b | 9.51 ± 0.09b |
| CCY10 | 4769.57 ± 164.57a | 4886.41 ± 137.82a | 178.89 ± 3.22a | 9.72 ± 0.06a |

**Notes.**

Values represent the mean ± standard deviation ($n = 4$). Different lowercase letters within the same column indicate significant differences among different continuous cropping years (Kruskal-Wallis, $P < 0.05$).

communities with a $P < 0.05$ significance threshold. The similarity of percentages analysis (SIMPER) was applied to identify the most affected genus between different continuous cropping years using the R package, Vegan (*Oksanen et al., 2013*). Procrustes included in the Vegan R package was used to compare the congruence of the microbiome and metabolism or functional profiles based on the $P$-values and goodness of fit ($m_2$). The co-occurrence network construction and parameter statistics were performed using Cytoscape v3.9.0 software (*Shannon et al., 2003*). The network was visualized using the interactive platform, Gephi (*Cherven, 2013*). In each network, the node size (i.e., degrees) was proportional to its number of connections; the thickness of each connection between two nodes (i.e., edges) was proportional to the Spearman's correlation coefficient ($|r| = 0.6–1$).

### Sequence accessions

The bacterial 16S rRNA gene sequencing data are publicly available in the NCBI Short Read Archive (SRA) under Bioproject accession No.: PRJNA759342.

## RESULTS

### Overall measures of diversity

We obtained 503,330 raw sequences and an average of 41,944 sequences per sample. After quality control and sample normalization, a total of 244,188 sequences were filtered and clustered into 6,745 unique ASVs (*Edgar, 2018*). The coverage of samples was >93% and the rarefaction curve of each sample approached a saturation plateau (Fig. S1), indicating that the current sequencing depth reflected the microbial composition. The Chao1, ACE, and PD indices were all significantly higher in CCY02 and CCY10 compared with CCY05 (all $P < 0.05$) (Table 1). The Shannon index was significantly higher in CCY10 when compared to CCY05, but no significant differences existed between CCY10 and CCY02, or CCY02 and CCY05, indicating that the bacterial diversity measured by the species richness with evenness had been reduced, but was not significant, during short continuous cropping years; after longer monocropping years, the whole bacterial diversity seemed to recover.

According to four different distance-based PCoA, we found that samples from the same cropping year clustered tightly, while samples from different years clearly separated based on both Jaccard (Fig. S2A) and Unweighted UniFrac PCoA (Fig. S2B), indicating that there clear differences existed in the absence or presence of ASVs among groups. Meanwhile, we found by Bray–Curtis PCoA (Fig. S2C) that, when considering the presence, absence, and abundance of ASVs, CCY05 obviously separated from CCY02 and CCY10. Additionally,

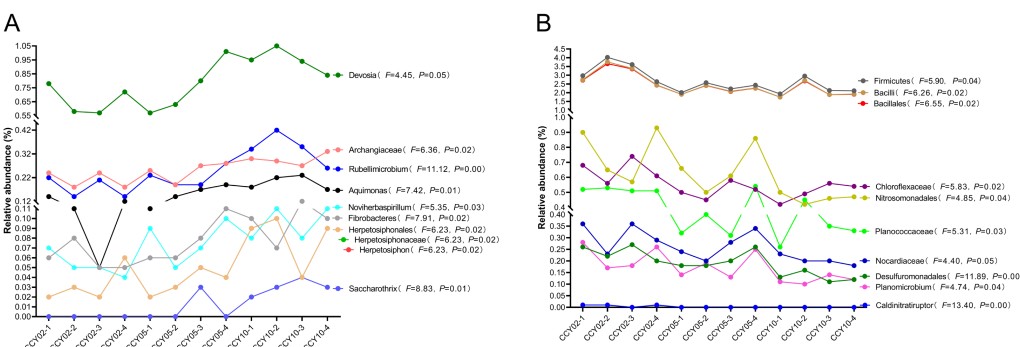

**Figure 1** **Total 20 taxa showed significant feedback continuously strengthening or weakening to the long-term monocropping of Tibetan barley.** (A) Ten taxa were increased, while (B) 10 taxa were decreased. The correlation between the relative abundance of each taxon and enhanced cropping years was further validated by ANOVA. $P < 0.05$ was considered significant.

there were no obvious differences detected among communities per the Weighted-Unifrac distances (Fig. S2D), which considered the presence, absence, abundance, and up-weighting differences in distant evolutionarily-related ASVs. Furthermore, the soil microbial compositions under different treatments were analyzed by Adonis statistical tests, which revealed significant dissimilarities among different cropping years (Table S1).

## Microbial community composition

Overall, Actinobacteria (average ± standard deviation (SD), 28.84 ± 1.50%) and Proteobacteria (26.88 ± 1.50%) were the dominant phyla across all rhizosphere soil samples. The genera with relative abundances >1% included *Blastococcus* (2.57 ± 0.30%), *Nocardioides* (2.55 ± 0.39%), *Sphingomonas* (2.44 ± 0.28%), *Bacillus* (1.09 ± 0.30%), and *Solirubrobacter* (1.02 ± 0.04%), but these genera did not significantly differ among continuous cropping years. After comparing the relative abundances of each known taxa among the three successive monoculture years, we found that six phyla, five classes, 11 orders, 23 families, and 47 genera exhibited a straight increasing trend over time, while five phyla, eight classes, 14 orders, 19 families, and 32 genera exhibited the opposite trend. A total of 20 taxa exhibited significant continuously strengthened or weakened tolerance to the long-term continuous cropping of Tibetan barley based on the ANOVA results (Fig. 1), but most of these taxa were low in abundance with a mean relative abundance <1%. For example, Fibrobacteres is an important phylum of cellulose-degrading bacteria (*Ransom-Jones et al., 2012*); the genus *Devosia* possesses bioremediation potential (*Talwar et al., 2020*); and the denitrifying bacteria *Noviherbaspirillum* (*Wu et al., 2021*) were significantly upregulated over time. The following significantly decreased over time: Firmicutes, which are plant growth-promoting bacilli (*Kumar, Khare & Dubey, 2012*); Bacillales, which have effective biological control and biodegradation potential (*Barathi et al., 2020*); Nitrosomonadales, which are related to sulfate and iron reduction; and Desulfuromonadales, which are an order capable of iron and sulfate reduction (*Wunder et al., 2021*).
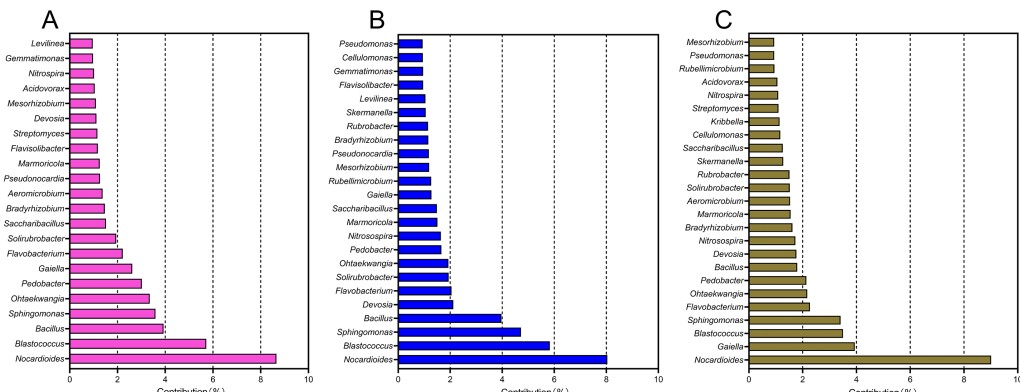

**Figure 2** **The relative contribution of main genera to the dissimilarity between communities of (A) CCY02 and CCY05, (B) CCY02 and CCY10, and (C) CCY05 and CCY10 by SIMPER analysis.** The main genera was listed when cumulative dissimilarity over 50% in each group.

A SIMPER analysis was used to detect the main genera that drove the compositional shifts of bacterial communities over time. The results showed that 22, 24, and 25 genera were responsible for >50% cumulative dissimilarity of the microbial community shifts between CCY02 and CCY05, CCY02 and CCY10, and CCY05 and CCY10, respectively (Fig. 2). Fourteen of these overlapping genera were mainly responsible for the differences between each pair of cropping years (Table S2), and the genera *Nocardioides* had the largest dissimilarity contribution.

## Core taxa among different cropping years

The consistency of crop type, planting strategies, and research site may lead to a cohort of rhizosphere soil-shared microbial communities from Tibetan barley, while continuous cropping over time could lead to year-unique taxa in the rhizosphere soil. In this study, the taxa that were simultaneously present in all samples were defined as core or shared taxa, and when their mean relative abundances (at the genus level) were >or <1%, they were defined as dominant or rare core taxa and subsequently classified at the phylum, class, order, family, and genus levels (*Gobet, Quince & Ramette, 2010*). At the genus level, the core shared microbiome consisted of nearly 50.87% of all taxa (taxonomic richness) and this percentage reached 60% at the phylum level, however, a proportion of unshared taxa seemed to be sample-specific (Fig. 3A). Surprisingly, the total relative abundances of the core taxa at the genus level occupied >98% of all sequences on average (Fig. 3A), while unshared taxa, or sample-unique taxa, only occupied 1.66 ± 0.31% of all sequences. A total of 164 core taxa were successfully annotated at the genus level and the relative abundances ranged from 0.02 ± 0.01% to 2.57 ± 0.61% (Table S3). The dominant shared genera among samples were *Blastococcus*, *Nocardioides*, *Sphingomonas*, *Bacillus*, and *Solirubrobacter* (Fig. 3B).

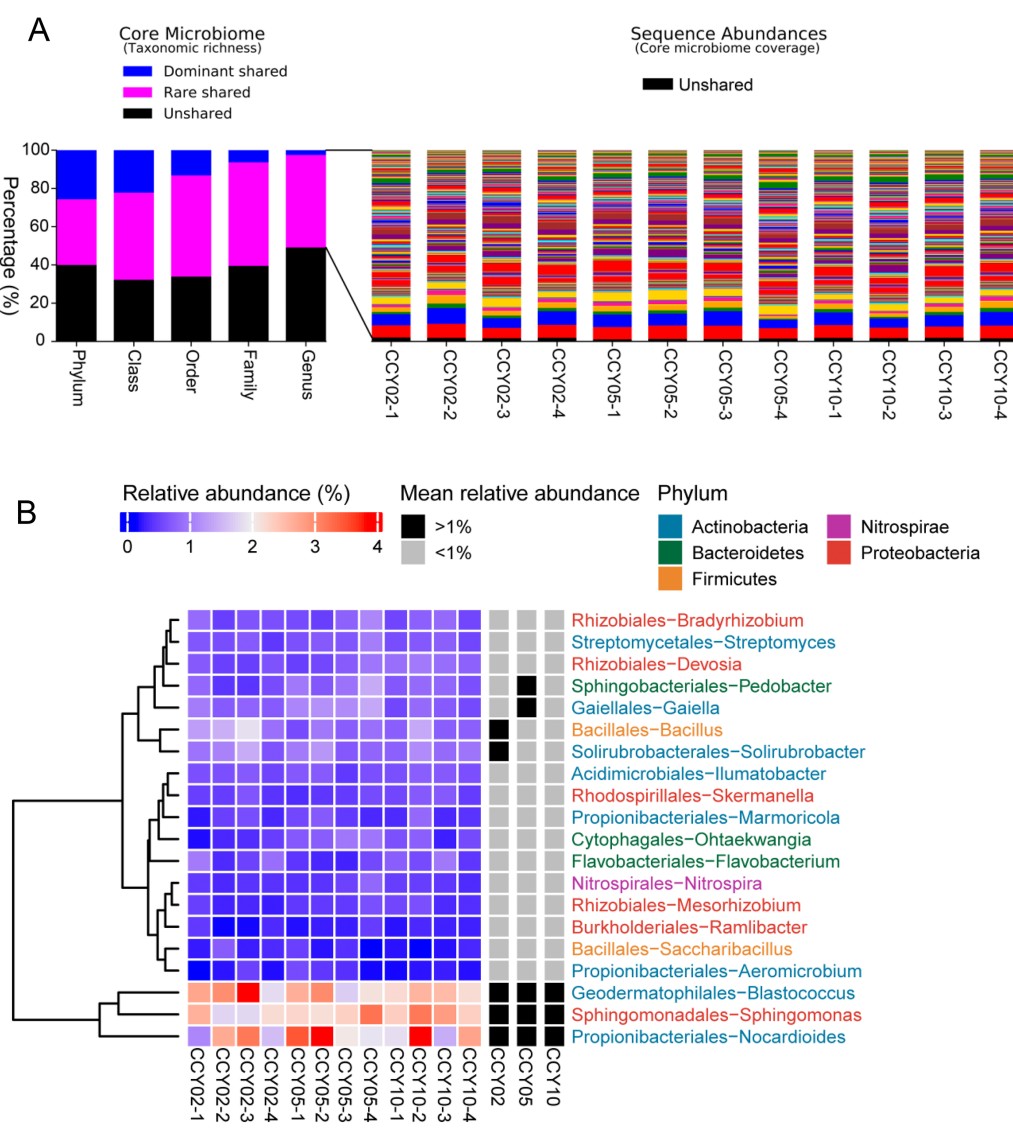

**Figure 3** **Richness, abundance, and identity of taxa are shared across all rhizosphere soil samples.** (A) Richness and sequence coverage of shared taxa in all the samples. (B) Heatmap of the top 20 shared genera among samples. Relative abundances were log-transformed and colored from blue to red to indicate high-to-low relative abundances. Shared genera were identified as dominant (>1% relative abundance) or rare (<1% relative abundance) in each group. The name of each genus is colored by phylum class.

## Bacterial functional genes

QMEC was implemented to determine the abundances of a wide spectrum of functional genes related to C-, N-, P-, and S-cycling (71 genes in total) in a high-throughput manner. A total of 60 genes were successfully detected by QMEC across 12 samples, including 17 genes related to C-degradation, 13 related to C-fixation, 19 related to N-cycling, 6 related to P-cycling, and 5 related to S-cycling (Table S4). Samples from CCY02 and CCY10 displayed a clear clustered pattern based on the principal components analysis (PCA), while samples from CCY05 were scattered (Fig. S3). The abundances of each gene in 12

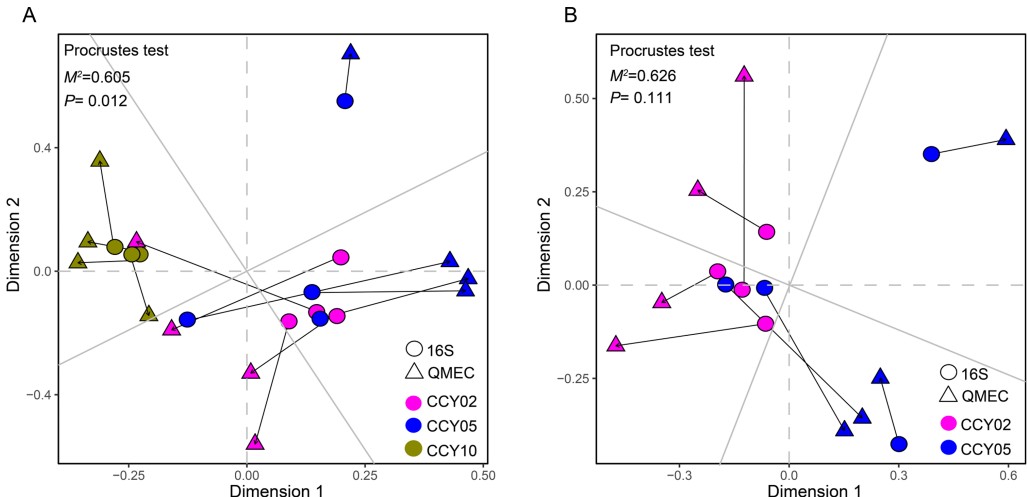

**Figure 4** Relationships between bacterial community structure and functional gene profiles from QMEC analysis during (A) long-term and (B) short-term of continuous monocropping years, respectively. The correlation was determined by Procrustes test and considered significant when $P < 0.05$.

samples are presented in Fig. S4. Briefly, we found that the genes involved in hemicellulose degradation (*abfA* and *xylA*) were most prevalent in C-degradation. The gene involved in the Wood–Ljungdahl pathway (*acsA*) and the gene involved in ammonization (*UreC*) were most abundant in C-fixation and N-cycling, respectively. In P-cycling, the genes involved in organic P mineralization (*phoD* and *phnK*) were the dominant genes across samples, while the gene related to S reduction (*apsA*) and the genes (*YedZ* and *SoxY*) related to S oxidation were the main genes in S-cycling.

We employed Z-values to compare differences in cycling genes among different cropping years (Table S5). The long-term continuous cropping of Tibetan barley significantly increased the abundances of genes related to C-degradation ($F = 9.25$, $P = 0.01$) and P-cycling ($F = 5.35$, $P = 0.03$) (Figs. S5A and S5B). The abundances of N-cycling genes increased from CCY02 to CCY05 and decreased by CCY10, but no significant differences were detected between groups (Fig. S5C). We also investigated the relationship between the abundances of functional genes and bacterial diversity (Shannon) (Table S6). Results revealed that only N-cycling significantly correlated with bacterial diversity ($r = -0.71$, $P = 0.01$).

## Relationships between taxa and functional genes

The Procrustes analysis showed that there were significant correlations between bacterial composition and functional gene profiles after long-term continuous cropping ($P = 0.012$) (Fig. 4A), but not after short-term continuous cropping ($P = 0.111$) (Fig. 4B). A co-occurrence network was constructed to visualize the relationships between the functional cycling genes and bacterial genera (Fig. 5). Only the interactions between the genera and functional genes were kept, of which 173 nodes and 481 edges (196 positive and 285 negative associations) were obtained. The network had good modularity (0.41) and

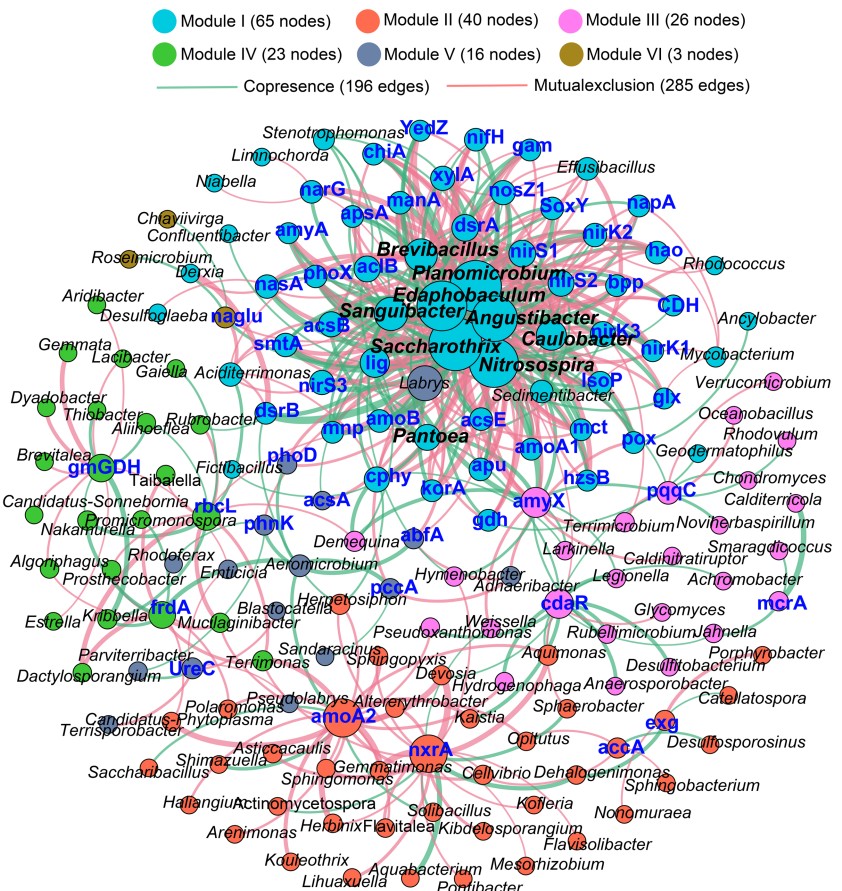

**Figure 5** **The co-occurrence network between the functional genes and all the bacterial genera appeared in all rhizosphere soil samples.** A connection indicates a strong ($|r|$ of > 0.6) and significant ($P$ of < 0.05) Spearman's correlation, while only a connection between genera and functional gene was kept. Red lines indicate negative correlations, while green lines indicate positive correlations. The network is colored by module, which clusters the tightly connected nodes. The size of each node is proportional to the number of connections: The thickness of each edge is proportional to the value of the Spearman correlation. The same node colors represent nodes belonging to the same modules. The font color blue represents the functional gene, while the font-weight represents the genera that clustered tightly within Module I with node degrees larger than 10.

contained 6 modules; each included tightly connected nodes. Module I, which was the largest, contained 65 nodes, including 43 functional genes and 22 genera; most of the C-, N-, P-, and S-cycling genes were tightly connected with a small group of genera. Nine key genera with a degree value >10 were identified, including *Saccharothrix*, *Planomicrobium*, *Edaphobaculum*, *Nitrosospira*, *Angustibacter*, *Sanguibacter*, *Brevibacillus*, *Caulobacter*, and *Pantoea*. Interestingly, *Saccharothrix*, *Edaphobaculum*, *Sanguibacter*, and *Pantoea* were positively connected with the functional genes, while *Planomicrobium*, *Nitrosospira*, *Angustibacter*, *Brevibacillus*, and *Caulobacter* were mainly negatively connected (Table S7).

We also examined the correlations between the cycling genes and 14 top contributors with the dissimilarities among different cropping years. Results revealed that *Ohtaekwangia*

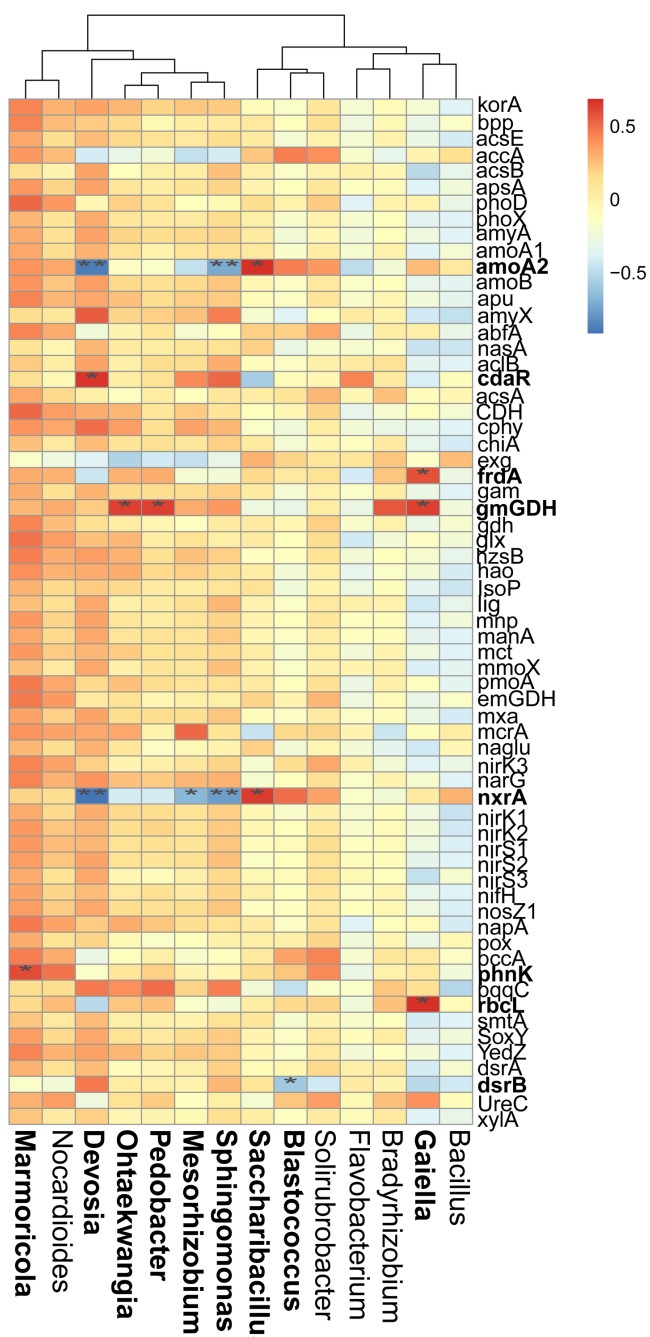

**Figure 6** **Correlations between the main overlapped genera (detected by SIMPER analysis) with the functional genes in rhizosphere soils.** Heatmap values ranged from +0.5 to −0.5. Values above/below zero represent positive/negative correlations. * $P < 0.05$ was considered significant, while ** $P < 0.01$ was considered extremely significant.

($r = 0.64$, $P = 0.02$), *Pedobacter* ($r = 0.63$, $P = 0.03$), and *Gaiella* ($r = 0.63$, $P = 0.03$) significantly positively correlated with *gmGDH*, which is related to C degradation (Fig. 6). With regard to C fixation, *Devosia* significantly positively correlated with *cdaR* ($r = 0.66$, $P = 0.02$), while *Gaiella* significantly positively correlated with *frdA* ($r = 0.58$, $P = 0.04$) and *rbcL* ($r = 0.68$, $P = 0.01$). For N-cycling, *Devosia* and *Sphingomonas* significantly negatively correlated with *amoA2* and *nxrA* ($P < 0.05$), while *Saccharibacillus* significantly positively correlated with these genes ($P < 0.05$). For P-cycling, *Blastococcus* significantly negatively correlated with *dsrB* ($r = -0.58$, $P = 0.04$), while *Marmoricola* significantly positively correlated with *phnK* ($r = 0.60$, $P = 0.04$). No significant correlations were detected between the five major categories of functional genes and 14 top genera (Fig. S6).

## Soil metabolic profiles

By LC-MS/MS-based non-targeted metabolomics, a total of 14,380 metabolites in the soils were obtained, in which, 822 metabolites were successfully annotated in the HMDB and KEGG databases. Based on the metabolic profiles, we implemented a PCA to uncover the different compositions of metabolites in the soil samples from different continuous cropping years. Results revealed that the first two principal components (PC1 and PC2) explained 62.93% of the total variance and samples from different groups obviously differed from each other (Fig. S7). The varying metabolic compositions among different groups were further confirmed by the heatmap analysis of the top 100 metabolites (Fig. S8).

VIP values in the PLS-DA model were calculated to examine the changes in the soil metabolites in greater detail. A total of 440 metabolites with VIP >1.0 and $P < 0.05$ (ANOVA) were considered significantly affected by the continuous cropping of Tibetan barley. Further linear regression analyses between the acquired metabolites and enhanced continuous cropping years were conducted. A total of 126 metabolites significantly responded to continuous cropping with an increase in 83 metabolites and decrease in 43 metabolites (Table S8). The changes in metabolites mainly occurred in lipids and lipid-like molecules, organic acids and derivatives, and organoheterocyclic compounds. Lipids and lipid-like molecules had the largest number of metabolites affected by continuous cropping. Moreover, nucleosides, nucleotides, and analogues were mainly upregulated over time, while alkaloids and derivatives were mainly downregulated.

## Relationships between taxa and metabolites

First, we applied Procrustes tests to depict the correlations between soil metabolic profiles and the bacterial community structure. Significant correlations were detected between specific metabolites and the bacterial community structure in rhizosphere soils during the long- ($P = 0.001$) and short-term continuous cropping of Tibetan barley ($P = 0.027$) (Fig. 7). To elucidate which microbial taxa were responsible for the changes in soil microbial metabolism, an interactive network linking the microbes with significant differences between groups (a total of 100 microbial taxa) and differential metabolites (a of total 126 metabolites) was constructed. Only the interactions between metabolites and microbial taxa were kept. The orders Desulfuromonadales (degree 57) and Nitrosomonadales (degree 19), families Archangiaceae (degree 34), Nocardiaceae (degree 23), and Sanguibacteraceae

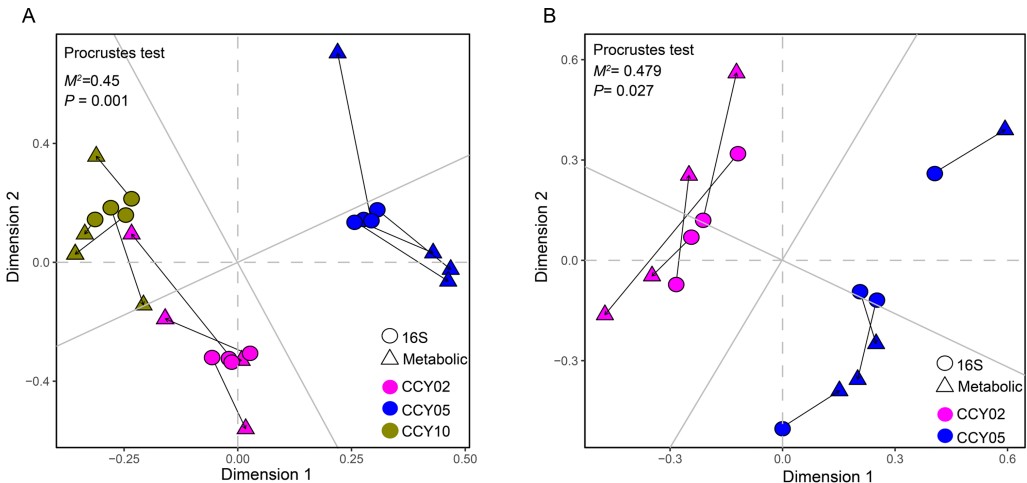

**Figure 7** **Relationships between bacterial community structure and Functional gene profiles from non-targeted metabolomic analysis during (A) long-term and (B) short-term of continuous monocropping years, respectively.** The correlation was determined by Procrustes test and considered significanlt when $P < 0.05$.

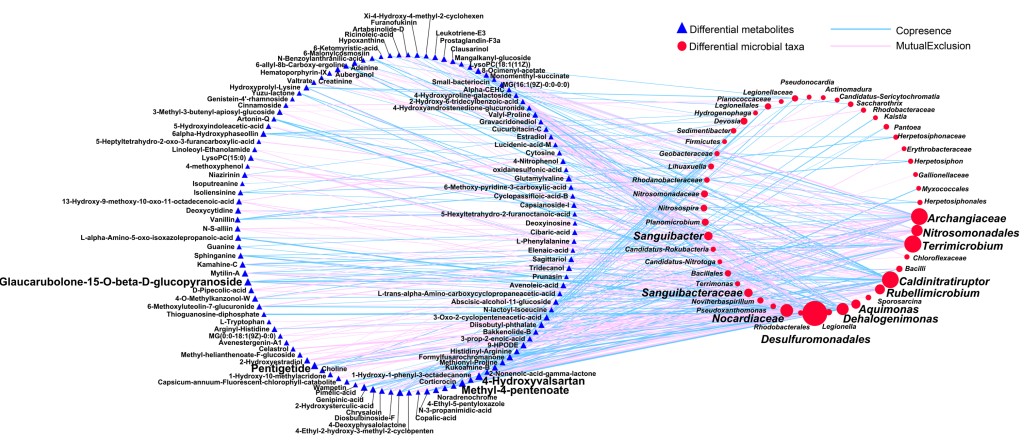

**Figure 8** **The network between the differential expressed metabolites and differentially abundant genera among different cropping years.** A connection indicates a very strong ($|r|$ of > 0.9) and significant ($P$ of < 0.01) Spearman's correlation, while only a connection between genera and functional gene was kept. Red lines indicate negative correlations, while green lines indicate positive correlations. The size of each node is proportional to the number of connections: The thickness of each edge is proportional to the value of the Spearman correlation.

(degree 11), and genera *Terrimicrobium* (degree 35), *Caldinitratiruptor* (degree 34), *Dehalogenimonas* (degree 21), *Rubellimicrobium* (degree 15), *Aquimonas* (degree 13), and *Sanguibacter* (degree 11) co-occurred most frequently with several differential metabolites, including 4-hydroxyvalsartan, methyl-4-pentanoate, pentigetide, and glaucarubolone-15-O-beta-D-glucopyranoside (Fig. 8). Of these metabolites, the frequently co-occurring ones were downregulated over time, except 4-hydroxyvalsartan.
## DISCUSSION

### Effects of continuous cropping on microbial community structure and diversity

Long-term continuous cropping may significantly alter bacterial community structure and diversity. For instance, *Yan et al. (2021)* reported that soil bacterial diversity significantly decreased (PD and Shannon) after the continuous cropping of *Nicotiana tabacum*. *Chen et al. (2020b)* found that the long-term consecutive monocropping of peanuts led to a general, albeit insignificant, loss in bacterial diversity. *Liu et al. (2020)* found that the bacterial diversity (PD and number of operational taxonomic units (OTUs)) was significantly higher after the long-term (13 year) continuous monoculture of soybeans, while no significant differences were detected after the short-term monoculture periods (3 and 5 years). *Zhao et al. (2020)* showed that bacterial richness (ACE) significantly decreased after 5–10 years of continuous cropping and recovered after 30 years, while bacterial diversity (Shannon) significantly increased after continuous cropping (5, 10, and 30 years). In the present study, bacterial diversity (Shannon) did not differ during short continuous cropping years, but bacterial richness (Chao1 and ACE) and PD significantly declined after short-term continuous cropping of Tibetan barley, which is consistent with the findings of *Yao et al. (2020)*, but surprisingly recovered after long-term continuous cropping (Table 1). This phenomenon may be due to the fact that all environmental filters require specific adaptation strategies for survival; stronger selection acting on bacteria often leads to a new cohort of microbiota that adapt to the environment (*Fierer, Bradford & Jackson, 2007*; *Meola, Lazzaro & Zeyer, 2014*). Even if the microbial composition is sensitive to environmental disturbances, the community may be resilient and stabilize over time (*Allison & Martiny, 2008*). However, we believe that the application of more powerful methods (*e.g.,* metagenome and metagenome binning) to deeply investigate the reasons for this phenomenon in future studies.

After 10 years of continuous cropping of Tibetan barley, Actinobacteria and Proteobacteria were the two main dominant phyla that were enriched in many rhizosphere soils after continuous cropping, such as in sugarcane and cotton (*Pang et al., 2021*; *Xi et al., 2019*). The widely distributed bacterial phyla, Actinobacteria, possess high proportions of CAZymes and exhibited a key important eco-physiological role in plant residue decomposition in a previous study (*Bao et al., 2021*). Another abundant phylum, Proteobacteria, plays a central role in the cycling of several key elements, including N- (*Conthe et al., 2018*), C- (*Chan et al., 2013*), and S-cycling (*Zhou et al., 2020*). In this study, a total of 20 low abundant taxa (<1%) significantly increased after continuous cropping, during which planting-promoting bacteria or bacteria with bioremediation abilities significantly decreased.

In the current study, we analyzed the top genera that contributed to the dissimilarities among microbial communities from different continuous cropping years. Of these, five were identified as the dominant shared genera across all soil samples, including *Blastococcus*, *Nocardioides*, *Sphingomonas*, *Bacillus*, and *Solirubrobacter* (Table S2). These genera have been frequently detected in the rhizosphere soils after the continuous cropping

of different crops, including maize (*Zhao et al., 2021*), cucumber (*Li et al., 2021*), cotton (*Han et al., 2017*), and peanuts (*Chen et al., 2018*). *Solirubrobacter* and *Blastococcus* exhibit heavy metal tolerance and advantages in stress resistance, indicating the promising potential for alleviating polluted soil ecosystems (*Hou et al., 2021*; *Wang et al., 2021*). A recent study suggested that *Sphingomonas* possesses multifaceted functions ranging from the remediation of environmental contamination to producing highly beneficial phytohormones that promote plant growth (*Asaf et al., 2020*). *Bacillus* has multiple medical, environmental, and industrial applications (*Khurana et al., 2020*). Hence, it is proposed that they may play important roles in the successive monoculture of Tibetan barley.

## Functional genes and their interactions with microbes

For the first time, QMEC was applied to determine the relative abundances of key element cycling genes after long-term continuous cropping. We found that overall bacterial composition significantly correlated with functional gene profiles (Fig. 4A), indicating that after long-term continuous cropping, and alterations in community functions were mainly affected by alterations in bacterial composition. However, after short-term continuous cropping (CCY02 and CCY05), the functional gene profiles and bacterial compositions had poor consistency (Fig. 4B). Thus, we proposed that the functional gene composition was not mainly mediated by changes in microbial composition after short-term continuous cropping. This may be a result of functional redundancy (*Rosenfeld, 2002*), which contributed to bacterial community tolerance and overall community functioning after the short-term continuous monoculture of Tibetan barley. However, significant fluctuations in the taxonomic composition after the long-term continuous monoculture may potentially reduce the stability of the community, which would thereby become more vulnerable to continuous disturbances (*Sheng et al., 2015*), ultimately affecting bacterial functional compositions.

Genes related to C-degradation and P-cycling increased over time (Table S5). Hemicellulose degradation (*abfA* and *xylA*) was most prevalent in C degradation, indicating the potential loss of soil C storage (*Chen et al., 2020a*). Organic P mineralization (*phoD* and *phnK*) increased after continuous cropping, playing an important role in soil P bioavailability. *Meyer et al. (2018)* indicated that microbes compensated for single nutrient deficiencies by accelerating P- or N-cycling and may have increased SOC turnover in co-limited subsoils with acquirable P reserves. Collectively, these results indicated that, along with the continuous cropping of Tibetan barley, the activity of hemicellulose degradation increased and accelerated C degradation, while microbes may have acquired greater C or N supplies by promoting organic P mineralization.

Furthermore, a set of genera were identified that showed tight correlations with most of the functional genes, indicating their functional potential in the cycling of key elements underlying the continuous monocropping of Tibetan barley (Fig. 5). In a previous study, *Nitrosospira* was the primary ammonia-oxidizing bacteria and was mainly responsible for nitrous oxide production (*Lourenço et al., 2018*). *Pantoea* fixes N or induces N uptake, thereby promoting N availability in plants (*Loiret et al., 2004*; *Singh et al., 2020*).

Additionally, *Brevibacillus* was reported to have desulfurization activity (*Nassar et al., 2013*).

A relevant study pointed out that the continuous cropping of Tibetan barley significantly increased the bacteria associated with chemo-heterotrophy, aromatic compound degradation, and nitrate reduction (*Yao et al., 2020*). This study showed that the dominant shared genera, *Sphingomonas*, exerted significant negative effects on nitrification gene abundances (*amoA* and *nxrA*), leading to the imbalance of nitrification and denitrification. Additionally, a previous study showed that *Sphingomonas* in vegetable systems utilized contaminants originating from pesticide residues as a growth and energy source (*Busse et al., 2003*).

## Soil metabolites and their interactions with microbes

Chemical dialog undergoing rhizosphere active zones promotes the interactions between plant roots and microbial communities (*Van Dam & Bouwmeester, 2016*). *Shen et al. (2020)* observed distinct differences in the metabolite composition over different continuous cropping seasons in tobacco. Our study explored the alterations in soil metabolic profiles, which were significantly mediated by soil microbial communities, indicating that certain metabolic pathways were altered for soil microbial communities to adapt to environmental stress (Fig. 7). Obvious differences in the patterns of metabolite compositions over time were observed (Figs. S7 and S8), in which the primary metabolites related to lipid and nucleoside metabolism were significantly upregulated, while secondary metabolites, such as alkaloids and derivatives, were significantly downregulated. A previous study pointed out that secondary metabolites greatly affect the microbial community, where some metabolites were antibiotic and pharmaceutically relevant, while others were involved in disease interactions (*Fox & Howlett, 2008*).

We found that the hub taxa Desulfuromonadales, which was reported to utilize different electron exchange pathways (*Rotaru et al., 2012*), mainly co-occurred with metabolites in either negative or positive correlations (Fig. 8), indicating potential important interactions between them. Meanwhile, a previous study suggested that positive co-occurrences indicated that the metabolites may have be mainly originated from microbes, while negative co-occurrences may represent the specific microbial consumption or degradation (*Devi et al., 2017*). These findings indicated that soil metabolomics can be used to assess the adaptations of soil microbial communities to continuous cropping strategies at the molecular level.

## CONCLUSIONS

Collectively, we analyzed the changing trends of bacterial composition and diversity after the continuous cropping of Tibetan barley using 16S high-throughput sequencing. After short-term continuous cropping, bacterial richness and PD significantly decreased, but recovered after long-term continuous cropping. We identified five dominant shared genera that were the main contributors to the dissimilarities among bacterial communities from different continuous cropping years, as uncovered by the core taxa and SIMPER analyses. Then, QMEC and untargeted metabolism were employed to determine the main functional

genes and soil metabolites, as well as their changing trends after different continuous cropping years. We further predicted the potential correlations between microbiota and metabolism, including functional genes and soil metabolites. Through this study, we gained further insights into the ecological roles of rhizosphere soil microorganisms underlying the continuous cropping of Tibetan barley.

## ACKNOWLEDGEMENTS

We thank LetPub for its linguistic assistance during the preparation of this manuscript.

### Funding

This work was supported by the Natural Science Planning Project of Qinghai Province (Grant nos. 2021-HZ-802), National Science Foundation of China (Grant nos. 32060447). The funders had no role in study design, data collection and analysis, decision to publish, or preparation of the manuscript.

### Grant Disclosures

The following grant information was disclosed by the authors:
Natural Science Planning Project of Qinghai Province: 2021-HZ-802.
National Science Foundation of China: 32060447.

### Competing Interests

The authors declare there are no competing interests.

### Author Contributions

- Yuan Zhao and Youhua Yao conceived and designed the experiments, authored or reviewed drafts of the paper, and approved the final draft.
- Hongyan Xu analyzed the data, prepared figures and/or tables, and approved the final draft.
- Zhanling Xie conceived and designed the experiments, analyzed the data, authored or reviewed drafts of the paper, and approved the final draft.
- Jing Guo, Zhifan Qi and Hongchen Jiang performed the experiments, prepared figures and/or tables, and approved the final draft.

### DNA Deposition

The following information was supplied regarding the deposition of DNA sequences:
The sequences are available at NCBI: PRJNA759342.

### Data Availability

The raw data of QMEC and non-targeted metabolomic data are available in the Supplemental Files. The raw data of 16S rRNA genes are available at NCBI SRA: PRJNA759342.

## Supplemental Information

Supplemental information for this article can be found online at http://dx.doi.org/10.7717/peerj.13254#supplemental-information.

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
