# Peer review of "Soil metabolomics and bacterial functional traits revealed the responses of rhizosphere soil bacterial community to long-term continuous cropping of Tibetan barley"

_PeerJ, doi:10.7717/peerj.13254_

## Round 0.1 · original submission · Minor Revisions

As you see from the attached reports, both reviewers requested only minor changes to the submitted manuscript, especially some clarifications and additional details regarding the statistical analysis were requested.

·

Basic reporting

The article is well written in English and the reading is clear, unambiguous with technically correct text. Besides, the manuscript includes professional standards of courtesy and expression.

I consider that introduction and background is sufficient to demonstrate how the work fits into the broader field of knowledge and include appropriate and relevant literature of the field.

Figures and Tables are relevant to the content of the article with sufficient resolution and content. However, I have these questions/suggestions:
- Table 1 (line 217): Does the data fulfill ANOVA assumptions of normality and homogeneity of variance? I suggest to use Kruskal-Wallis test instead of ANOVA for alpha diversity indices.
- Figure S2: please identify in the text what is A and B, in order to appropriately describe and label the figure.

The structure of the article follows an acceptable format and all appropriate raw data have been made available in NCBI-SRA.

Experimental design

Here, I expose my manuscript suggestions:

line 120: could you specify the dimension of each plot? could you clarify the total samples collected? Did you collect 15 rhizosphere samples from each plot or from each five sites within a plot? The sampling procedure and the number of samples you have is a bit confusing, please clarify it by indicating the total sample you have and how they are distributed within each plot.

line 133: despite referring a previous work from Yang et al. 2019, I suggest to indicate briefly the process you followed for PCR amplification and purification.

line 134: what chemistry did you use for MiSeq sequencing? paired-end? single end? V3; 2x300 pb? please indicate it.

line 137: if you have 15 rhizosphere samples, why did you select 12 for the different analyses performed?

line 184: did you consider additional quality steps to depurate more accurately the raw reads obtained?

line 191: if you considered Faith’s phylogenetic diversity in alpha diversity indices, why did you not considered Weighted or Unweighted UniFrac distances in beta diversity?

line 199: Did you build the network with the total ASVs obtained? I recommend to apply a prevalence to build a robust network. Besides, when you analyze a co-ocurrence network you need to consider other correlations apart from Spearman, as well as you have to determine similarities and distances values as you can see in the Figure 1 of the manuscript by Faust et al., 2012 (Microbial Co-occurrence Relationships in the Human Microbiome).

Validity of the findings

I would like to congrats the authors for this comprehensive work and I would like to highlight that this study provides new insights about the interaction of soil metabolomics and bacterial functional traits in the rhizosphere soil bacterial community to long-term continuous cropping of Tibetan barley. Besides, the authors include a wide range of relevant analyses, such as network associations between microbiota and metabolisim that result very interesting to dig deeper in future experiments.

Additional comments

Please, check and rewrite the name of the genera in italics within the manuscript.

I suggest taking into account fungal communities in future experiments to understand their role in soil rhizosphere with similar analysis you performed.

Finally, I recommend to select larger samples to provide more conclusive results in future experiments because the results may be biased by the low number of replicates you selected.

Reviewer 2 ·

Basic reporting

Continuous cropping seriously affects crop yield. The authors analyzed the bacterial composition, microbial functional traits and soil metabolites of the rhizosphere soil collected from continuously cropped Tibetan barley for 2, 5, and 10 years. I feel this paper have important significance on barley continuous cropping.

Experimental design

he authors described 15 rhizosphere soil samples were collected in Line 122, but in Line137, twelve rhizosphere soil samples were collected. Please check it.

Validity of the findings

The Chao1, ACE, and PD indices were all significantly higher in CCY02 and CCY10 compared with CCY05. This result seems unreasonable. It means that soil condition will become better when continu-ous mono-cropping years is more than 10 years?

Additional comments

1. In Abstract section: “samples from the same mono-cropped year”. The meaning of this sentence is not clear. “continuous momo-cropping year” may be better understood.
2. Line 363: Soil bacterial diversity may significantly decrease, have no change, or fluctuate after long-term continuous cropping. I think this sentence has something wrong. In general, soil bacterial diversity was decreased with the increase of continuous mono-cropping years. No change is extremely unlikely to happen. If this description was quoted from the reference, the result of this reference is questionable.

---

## Round 0.2 · accepted · Accept

All the reviewer comments were properly addressed. I do not have any more comments and hence accept the manuscript for publication.

·

Basic reporting

In my opinion, I consider the manuscript can be published in PeerJ.

Experimental design

The authors have made the revisions to the manuscript and they addressed all my concerns and suggestions.

Validity of the findings

This study provides new insights about the interaction of soil metabolomics and bacterial functional traits in the rhizosphere soil bacterial community to long-term continuous cropping of Tibetan barley.

Reviewer 2 ·

Basic reporting

No comment.

Experimental design

No comment.

Validity of the findings

No comment.

Additional comments

No comment.